# Motor Chunking in Internally Guided Sequencing

**DOI:** 10.3390/brainsci11030292

**Published:** 2021-02-26

**Authors:** Krishn Bera, Anuj Shukla, Raju S. Bapi

**Affiliations:** Cognitive Science Lab, Kohli Research Center on Intelligent Systems, International Institute of Information Technology, Hyderabad 500032, India; krishn.bera@research.iiit.ac.in (K.B.); anuj.shukla@research.iiit.ac.in (A.S.)

**Keywords:** learning, motor sequence learning, motor chunking, internally guided sequencing, grid navigation tasks

## Abstract

Motor skill learning involves the acquisition of sequential motor movements with practice. Studies have shown that we learn to execute these sequences efficiently by chaining several elementary actions in sub-sequences called *motor chunks*. Several experimental paradigms, such as serial reaction task, discrete sequence production, and m × n task, have investigated motor chunking in *externally specified* sequencing where the environment or task paradigm provides the sequence of stimuli, i.e., the responses are stimulus driven. In this study, we examine motor chunking in a class of more realistic motor tasks that involve *internally guided* sequencing where the sequence of motor actions is self-generated or internally specified. We employ a grid-navigation task as an exemplar of internally guided sequencing to investigate practice-driven performance improvements due to motor chunking. The participants performed the grid-sailing task (GST) (Fermin et al., 2010), which required navigating (by executing sequential keypresses) a 10 × 10 grid from start to goal position while using a particular type of key mapping between the three cursor movement directions and the three keyboard buttons. We provide empirical evidence for motor chunking in grid-navigation tasks by showing the emergence of subject-specific, unique temporal patterns in response times. Our findings show spontaneous chunking without pre-specified or externally guided structures while replicating the earlier results with a less constrained, internally guided sequencing paradigm.

## 1. Introduction

Skill learning refers to the acquisition of the ability to perform a task with ease. It is a cornerstone of human cognitive ability that enables us to master various tasks that we deal with daily. It is a natural behavioral phenomenon observed while performing different tasks spanning a range of difficulty and timescales. For example, it takes a couple of hours or days to master simple sequencing tasks on the keyboard, whereas it can take months or years to master complex motor skills such as driving or ice-skating. Previous studies in developmental psychology have also tried to investigate the mechanisms underlying motor actions over longer timeframes [1,2,3,4,5]. Motor tasks typically require coordinating between planning, selection and execution of motor actions to facilitate skillful behavior [6,7]. Motor skill learning refers to the acquisition of skillful motor behaviors such that they are executed efficiently with practice [8,9,10,11,12]. Motor learning is a function of two specific categories of skill acquisition–motor adaptation and motor chunking. Motor adaptation is the context-specific, error-driven acquisition of locomotor patterns, which involves adjusting to changes in sensory input or motor output characteristics [13,14]. A practical example of motor adaptation is learning to use a mouse pointer with different sensitivities. With changes in mouse sensitivities, we adaptively learn to map our hand movements to mouse pointer movement changes. Motor chunking involves the consolidation of certain movement elements into clusters that allow efficient execution of the multi-element sequence [15,16]. With repeated rehearsals, the individual elements are consolidated into a sequence of quick motor actions. For example, motor chunking is observed when we learn the swift execution of keypresses while playing a videogame. Complex action sequences in the game are broken down into multiple simple components (e.g., “Jump-forward-and-Shoot” or “Move-right-Duck-Jump-forward”).

Much of the early interest in motor chunking focused on how the repeated execution of visuomotor sequences leads to overall performance improvement. Studies have shown that the inter-response intervals within certain sub-sequences decrease with practice compared to those in-between these sub-sequences [17,18,19]. It leads to the emergence of distinct clusters of motor movements (called motor chunks), which facilitates efficient sequence execution [20,21,22]. The chunks segment the motor sequences into smaller representational clusters, reflecting integrated sequence representations [18,19]. With substantial practice, the temporal patterns become more prominent and the chunks can be identified more distinctively [15,16,23]. The benefit of such segmentation of the action sequences is that it reduces memory load during sequence execution [17,24,25,26,27]. Each motor chunk is stored as a single memory representation and the entire chunk is loaded at once into the motor buffer for execution. This results in “automatic” control of sequential movements with reduced cognitive demand [28]. Previous studies have also shown that motor chunking is not merely an effect of rhythm consolidation in sequential motor tasks. The time or memory resource constraints can also inhibit motor chunk loading, leading to performance degradation [29,30].

The emergence of similar cluster patterns has been observed in a variety of motor sequence learning tasks. In the explicit domain, motor chunking has been shown in paradigms such as discrete sequence production task (DSP) and m × n task [15,16,28,31,32,33,34]. While chunking has been consistently reported during explicit learning, some studies (e.g., serial reaction task—SRT) have also provided evidence for cluster pattern of motor sequence performance during implicit learning of visuomotor sequences [35,36,37,38]. In SRT and DSP tasks, the participants execute a sequence of keypresses in response to a fixed set of visual stimuli. The stimuli are organized in a set of trials and are presented repeatedly during the experiment. The keypress responses to the set of visual cues are recorded on a button-box where the keys are visuo-spatially compatible with the stimulus presentation. In the m × n task, the visual stimuli are presented as *m* illuminated squares on a 3 × 3 grid. The participants discover and learn the correct order of the sequence of *m* keypresses (called a set) by trial and error. In all such paradigms, the set of visual cues that guide the sequencing behavior is fixed and pre-determined by experimental design. The sequence of motor actions to be performed in such tasks is contingent on externally specified visual stimuli and is not dependent on the participant’s plan or volition. Therefore, such canonical paradigms, which have been typically employed to investigate sequence learning and motor chunking, involve *externally specified* sequencing.

While these experimental paradigms are useful for studying the typical behavioral phenomena in sequence learning in laboratory settings, they are not representative of many real-life motor tasks. One crucial aspect in which they differ is that many real-life motor skills are *internally guided*. The sequence of motor actions in such tasks is self-initiated (generated internally) by choice or internal plan. The sequence of motor actions is not prescribed externally by the environment or experiment. While visual cues might help the agent determine the environment’s state, it does not convey the sequence of motor actions to be executed. The characteristic difference between externally guided and internally guided sequencing is that, in the latter, the agent needs to plan or choose the sequence of motor actions by itself. A practical example of an internally guided task is composing a music piece on a keyboard. It involves sequential keypresses to produce the musical notes, but the sequence of keypresses is not externally specified. The sequence of motor actions is internally planned according to the rhythm of the music being composed. Similarly, a tracing task is an example of the externally guided paradigm, whereas a drawing task belongs to the internally guided paradigm. The performance in such internally guided sequencing depends on the dexterity of executing the motor actions and the ability to program the sequence of future actions. Consequently, internally guided behavior involves greater demand for memory and planning. The neuro-imaging studies have confirmed these differential activations in the brain [39,40,41,42].

In externally guided sequencing, learning is facilitated by associations between the stimuli and the responses using simple rules [43]. Repeated execution of these sequences results in sensorimotor learning through the stimulus–response–effect (S–R–E) chain [44]. On the other hand, internally guided sequencing is executed as a chain of response–effect (R–E) contingencies as bindings form between the executed movements and the resulting sensory effects [44]. In light of this differentiation, most previous studies examine motor chunking in externally guided sequencing. An open problem yet to be investigated is whether chunking plays a role in motor learning in internally guided paradigms. Our present study investigates the role of chunking in practice-driven performance improvements in internally guided sequencing.

Our study proposes a novel usage of the *grid-sailing task* (GST) [45,46] as a canonical paradigm to investigate internally guided sequence learning. We hypothesize the role of chunking in practice-driven performance improvements in GST. The task required participants to discover an optimal path to the goal position (via a sub-goal) using the learned KM. They executed the same trajectory in all the subsequent trials to consolidate the learning. First, we show overall learning, as reflected in improved execution times in successful trials with practice. Then, we examine the underlying temporal patterns of keypress response times to show that the sequential keypresses are organized in subsequences or chunks, facilitating efficient behavior. We show how these clusters consolidate into fewer, larger chunks with practice in internally guided sequencing. The significance of our study lies in probing the role of chunking during motor learning in real-life paradigms. The externally guided paradigms, such as SRT or DSP, involve “passive” motor learning behaviors, which are guided by external stimuli and so findings from these studies cannot be generalized to most practical motor tasks. Using canonical paradigms, such as grid-navigation, our study highlights how motor learning can be investigated in more practical tasks.

## 2. Methods

The repeated sequential execution of keypresses in grid-navigation tasks amounts to sequence learning in an internally guided fashion. These tasks involve navigating a cursor from the start position to the goal position on the grid. The possible cursor movements are associated with particular keyboard buttons in a one-to-one correspondence. Each individual path or trajectory from start to goal position constitutes a novel sequence of keypresses. The optimal trajectory to the goal is dependent on other task specifications such as possible agent movements and reward schema. To complete the trial successfully, the participants can choose to reach the goal position using any possible optimal trajectories. The repeated execution of these trajectories results in learning a self-generated, voluntary sequence of keypresses. This behavior during the sequential execution of motor actions provides us with rich insights into how we become increasingly proficient in internally guided tasks. Therefore, we employ an adapted version of grid-sailing task [46] to investigate chunking behavior in internally guided sequencing. Moreover, GST is a simple task and so the dissociation of the sequence-specific motor learning is relatively easier when controlling for the trajectories executed. It is also a flexible canonical paradigm that can be altered on factors, such as key mapping, start-goal position, size of the grid and reward schema, to facilitate this dissociation and flexibly generate multiple variations of the task.

### 2.1. Participants

Fifteen right-handed participants (7 women and 8 men) between ages 18 to 27 years (mean = 21.53; SD = 2.79) performed the experiment for partial course credits. All participants were healthy with normal or corrected-to-normal vision. The experiment was approved by the Institute Review Board, IIIT-Hyderabad, India. The participants gave informed consent before the study. Two participants did not complete the experiment as they were unable to recall the first optimal trajectory that they traversed. The data from their attempts were excluded for all purposes. The data from the remaining thirteen participants were used for all analysis purposes.

### 2.2. Apparatus

The participants were seated on a chair facing a high-resolution 24” computer screen placed approximately two feet away. The responses were recorded using a conventional computer keyboard. The participants used the right, middle and index fingers to press the numpad buttons “4”, “5” and “6”, respectively. Other keys were removed to prevent meddling in response selection. Custom-made programs were written using Python3 and PyGame (Python Game Development) for stimulus presentation and data recording.

### 2.3. Task Paradigm

The subjects were given verbal instructions about the task rules before the session started. Each trial began with the presentation of a 10 × 10 grid with a red fixation on the center of the screen. On pressing the space button, after a random delay of 500–1000 ms, the trial would begin with the start position marked as a green tile, the sub-goal position marked as a red tile and the goal position marked as a blue tile. The cursor was shown as a black triangle, initially placed in the starting position. The participants were given 9 s to solve each trial, and this duration was not explicitly conveyed to them. We computed this to be an ideal adjusted trial duration based on the average length of optimal trajectories and the mean execution time in the original GST study [46]. During the trial duration, participants executed sequential keypresses to navigate the cursor from the starting position to the goal position via the sub-goal. The sub-goal was introduced to help participants in navigating to the goal position. To complete the trial, the participants must reach the goal position only after visiting the sub-goal position. The possible cursor-movement directions were defined by the key mapping (KM) (see Figure 1B). Apart from the possible number of movement directions, no other information about the key mapping was conveyed to the participants. The task required participants to explore the possible KM movement directions and the corresponding key associations by trial-and-error.

The participants were instructed to achieve a maximum score (of 100 points) while executing each trial as quickly as possible. A maximum of 100 points was awarded when the participants traversed an optimal path to reach the goal position. A minimum-steps trajectory from start to goal via the sub-goal position is considered an optimal path. Suppose a non-optimal path was traversed, a penalty of −5 points incurred for every excess move. In case the participant tried to perform an infeasible move, such as moving out of the grid, the cursor stayed there, but the action increased the move count. If the participant failed to reach the goal position in the given time duration, 0 points were awarded for that trial. At the end of each trial, the performance feedback was presented for 2 s, following which the fixation screen signaled the beginning of a new trial. In the center of the feedback screen, the performance feedback was presented as two numbers: the number of moves in the traversed trajectory and the trial reward score. A trial illustration is shown in Figure 1C.

The trajectory was controlled for all subsequent trials to investigate the practice-driven performance improvements with the repeated execution of sequences. The pair of start-goal (SG) positions and the sub-goal position remained constant throughout the experiment. The participants were asked to remember the first optimal path (minimum-steps trajectory; reward score = 100) traversed on the grid. Once they discover an optimal path, the trajectory traversed in that particular trial was locked in the program. In all subsequent trials, the participant must repeatedly traverse the same path to reach the goal and complete the trial successfully (reward score = 100). Any deviation from the locked path will award the participant zero points and the trial will be deemed unsuccessful. The experiment ran until subjects successfully performed 60 trials while repeatedly traversing the optimal path that they first discovered. The participants were given a rest block after every 20 trials to minimize the effects of muscle fatigue on task performance.

### 2.4. Behavioral Measures

The number of moves in the traversed trajectory, reward obtained, reaction time and execution time were the performance variables recorded for each trial. Individual keypress response times (RTs) were also recorded for key-level analysis purposes. Reaction time is defined as the time interval between the onset of stimuli and the first keypress. Execution time is computed as the difference between the keypress time of the last and the first response. For analysis purposes, the trials were classified into two categories (1) successful trials—trials with perfect reward score (equal to 100), and (2) error trials—trials with an imperfect reward score (not equal to 100). Only successful trials were included for all analysis purposes.

## 3. Results

### 3.1. Learning in GST

Participants performed the grid-navigation task and the behavioral measures on each trial were recorded. On average, each participant attempted 87 trials to complete 60 successful trials. The error trials constituted 30.6% of all trials. To examine how the learning evolves with GST, we plotted mean error rates across all participants. The error rate is computed as the number of error trials attempted to complete each successful trial. In Figure 2A, we observe that most of the error trials occur during the initial trials of the task when the participants are learning to use the key map to find an optimal path to the goal position. The error rates drastically decrease after participants discover the first optimal trajectory.

On plotting mean execution times for successful trials across participants, we see a decreasing trend in the plot (see Figure 2B). The decrease in execution times over trials can be attributed to performance improvements due to learning. With practice, the participants learned the KM and utilized it to plan an optimal trajectory to the goal position. Further performance improvements followed as they learned to execute the trajectory sequences swiftly. The law of practice effect on learning is evident by examining the initial and last trials. In the beginning, the participants took a longer mean execution time of 6247 ms (SD = 1303) to complete the first trial successfully. After substantial practice, the mean execution time on the last trial was significantly lower at 3822 ms (SD = 766). The execution time improvements suggest that the participants learned to navigate and execute sequential keypresses efficiently with practice. A non-parametric Friedman test of differences among repeated measures (within-subjects) rendered a significant effect of trials on the execution time ($\chi^{2}\left(59\right)=291.198,\;p<0.001$).

### 3.2. Motor Chunking in GST

The decrease in execution time over trials is illustrative of the performance improvement due to skill learning. The improvement in execution time was observed even when participants repeatedly executed the locked optimal trajectory (i.e., reward = 100). While the reward certainly guided the planning of motor actions during initial trials by providing feedback on the optimality of the trajectory, it becomes inconsequential to motor performance once the optimal trajectory is discovered. At this point, the reward only indicates whether the same optimal trajectory was followed in the subsequent trials. Such binarized reward feedback incentivizes the participants to repeatedly execute the same internally guided trajectory throughout the remaining experiment. This enables us to probe how spontaneous grouping structure emerges in sequence execution as the motor performance is fine-tuned and optimized. We hypothesize the role of motor chunking in practice-driven performance improvements. We identify the motor chunks based on changes in temporal patterns of keypress RTs.

#### 3.2.1. Identifying Chunk Patterns from Keypress RTs

The chunks were identified using Wilcoxon signed-rank tests between successive keypress response times (RTs). Studies have shown that the initial element in a chunk typically exhibits a slower behavior because of the RT cost of initializing and loading the motor chunk [3,9,14]. Therefore, the keypress RT for the first element of the chunk is significantly different from the next element. In agreement with this argument, if the $n^{th}$ and ${\left(n+1\right)}^{st}$ keypress RTs are significantly different and the ${\left(n+1\right)}^{st}$ keypress RT is less than $n^{th}$ keypress RT, both will belong to the same chunk. Additionally, in case the ${\left(n+1\right)}^{st}$ keypress RT is significantly higher than the $n^{th}$ keypress RT, both keypresses will not belong to the same chunk. All successive elements with non-significant keypress RT differences belong to the same chunk. To be considered as a chunk, the segment should have at least two keypress elements. Therefore, the three operating rules to identify chunks are (i) the initial element in a chunk is typically characterized by a significantly higher RT than the following keypresses; (ii) only successive elements with a statistically insignificant difference or monotonically decreasing keypress RTs are appended to the current chunk; (iii) a significant increase in keypress RT denotes the beginning of a new chunk. Figure 3 shows motor chunks (marked by brackets) in four representative subjects in early and late practice phases. The first ten trials (trials 1–10) belong to the early practice phase, whereas the last ten trials (trials 50–60) belong to the late practice phase.

#### 3.2.2. Re-Organization of Action Sequences with Practice

To corroborate our findings, we also examine the emergence and re-organization of motor chunks from the early practice phase to the late practice phase. Previous studies have shown that the motor chunking in sequence learning tasks evolves with practice-induced changes in temporal patterns of execution [8,9]. To check if the performance in the early phase is different from that in the late phase, we computed the mean keypress RTs in each phase by averaging all the keypress RTs (1–16) in each phase (10 trials) across all the subjects (*N* = 13). A paired t-test between mean keypress RTs for early (mean = 358 ms; SD = 68) and late (mean = 257; SD = 49) phase showed a significant difference (t(12)=11.435, p<0.001).

As the temporal patterns of sequence execution dynamically evolve with practice, the chunks re-organize with repeated concatenation and segmentation. Table 1 records changes in chunk features such as the number of chunks formed and the average length of chunks over the course of practice in four representative subjects. For example, subject N.J. executed the entire sequence in five segments during the early phase (red) and three segments during the late phase (blue). The average chunk length increased from 2.80 to 5.33. The area between the two lines (red and blue) in Figure 3 indicates performance gains with the re-organization of the chunks. A general increasing and decreasing trend were found across all participants from the early to late phase for the average length of the chunk and the number of chunks, respectively (see Figure 4). The average number of chunks across subjects in the early and late phase were 3.923 and 3.154, respectively. As normality assumptions were violated, we used a Wilcoxon signed-rank test, which suggests that the decrease in the number of chunks was significant (Z=2.5, p=0.026). The average chunk length across subjects increased from 4.153 in the early phase to 5.263 in the late phase. A Wilcoxon signed-rank test suggests that the increase in chunk length was also significant (Z=6.5,p=0.018).

## 4. Discussion

Motor chunking has been extensively studied in externally guided tasks in both implicit and explicit domains. Not many studies have investigated chunking in internally guided sequencing tasks. To the best of our knowledge, this is the first study investigating motor chunking in internally guided paradigms. Using grid-navigation as an exemplar paradigm, we hypothesized the role of motor chunking in practice-driven performance improvements in internally guided sequencing. First, we analyzed the effect of trials on execution time to show trajectory-specific motor learning in GST. Then, we analyzed the temporal patterns of keypress RTs to provide evidence for motor chunking. Distinct clusters of swiftly executed successive elements emerge with practice. We found that the keypress RTs are different in the early and late practice phase. We observed a significant improvement in keypress RTs in the late phase due to chunk consolidation. We further showed how the chunk characteristics, such as chunk length and number, evolve to facilitate efficient execution. The number of chunks decreased as the length of chunks increased from the early practice phase to the late practice phase (see Table 1 and Figure 4). With practice, smaller chunks coalesce to form bigger chunks to promote simpler integrated sequence representations that can be executed efficiently. We also observed that the chunk boundaries did not simply correspond to switching between the keys, reiterating that chunking is not merely functional. We also found that in the early phase, the participants were slower on the sub-goal, and hence, the sub-goal marked a chunk boundary. As the chunks evolved with practice, the sub-goal was consequently executed as a medial keypress within the chunk for most participants. It is indicative of the re-organization of the chunks, which happens to facilitate efficient execution. The chunking behavior was found universally across all subjects. The chunking pattern in the sequence was individual or subject-specific.

Our study corroborates findings from other studies on chunking and internally guided tasks. The learning in GST is a function of both cognitive and motor components. While cognitive learning relates to the ability to navigate the grid using the acquired key map, motor learning involves acquiring skillful sequencing behavior. The motor learning in GST is characterized by the improving dexterity of executing the sequences repeatedly. In line with previous work in externally guided tasks, we show that the sequence learning in GST-like internally guided tasks is facilitated by segmenting the sequence into motor chunks [12,13]. Our findings also corroborate with the computational account of learning proposed in [29]. The chunking-driven efficiency can be attributed to the model-free, memory-based strategy that the participants use to reproduce pre-learned action sequences after extensive practice. Previous studies have identified chunking as a cost-effective learning strategy that reduces overall computational complexity while maintaining efficiency [11]. Neuro-imaging studies investigating GST and cued-sequence production tasks provide evidence for the shared neural underpinnings of motor-memory guided actions and chunking (segmentation and concatenation) processes. The supplementary motor area, putamen and anterior cerebellum areas are involved in GST in the late phase when the actions are habitual, automatized and driven by model-free learning [30]. The studies have shown the role of sensorimotor putamen, frontoparietal network and pre-supplementary motor area during chunking behavior in sequencing tasks [3,10]. Given these complementary findings, future work can investigate if a computational framework of chunking can be conceptualized with the joint modeling of response times and decisions in internally guided sequencing tasks.

While our study replicates previous findings from the chunking literature on a less constrained, internally guided sequencing paradigm, it is important to situate the behavioral phenomena in chunking in the context of the learning processes involved in internally guided paradigms. In contrast with externally guided actions, internally guided or intention-based actions involve higher-order cognitive processes such as planning, memory and decision-making. Therefore, it is crucial to understand whether internally guided and externally guided sequence learning have similar underlying mechanisms in light of this differentiation. The sequence learning in externally guided paradigms has long been explained with multiple single-level accounts (see [47] for a review) of response location associations, perceptual and response effect learning. Subsequent studies can probe if similar learning mechanisms are at play during internally guided sequencing. The internally guided tasks will allow us to disentangle and understand the contributions of perceptual and response–effect learning. In line with some previous results [48], we speculate that serial learning in internally guided tasks involves voluntary action control mechanisms characterized by the acquisition of response–effect contingencies. In the ideomotor framework of action and perception, voluntary action goals have been shown to influence internally guided sequencing [49,50,51]. Based on previous neuroimaging studies [40,42,52], we anticipate the role of basal ganglia, pre-supplementary motor cortex and dorsolateral prefrontal cortex in chunking in internally guided sequencing tasks.

Previous studies [21,53] have shown that chunking occurs as a hierarchical organization of motor action sequences in externally guided visuomotor sequencing. Future work can experimentally probe theoretical questions about hierarchical re-organization—i.e., how chunks are formed and integrated into a multi-level structure. Subsequent studies can also use other methods of identifying and segmenting chunks in internally guided sequencing. For example, [54] proposes a Bayesian algorithm that identifies chunks based on response times, errors and their correlations. The study in [23] uses a multi-trial community detection approach after constructing the sequence network.

## 5. Conclusions

Our study provides evidence for practice-driven performance improvements in GST due to motor chunking. In contrast with previous studies, we show chunking in a more realistic and practical motor paradigm called internally guided sequencing tasks. Our findings show evidence for spontaneous chunking without pre-specified or externally guided structures while replicating the earlier results with a less constrained experimental paradigm. We also show how the chunks evolve and re-organize during various phases of the practice. Our findings call for a renewed interest in investigating motor chunking and other skill-learning aspects of sequencing in internally guided tasks.

## Figures and Tables

**Figure 1 brainsci-11-00292-f001:**
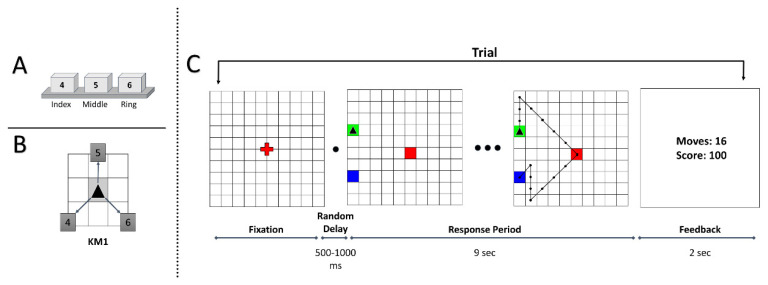
(**A**) Numpad keys and the respective hand fingers. (**B**) Key mapping (KM) used in the experiment. The marked arrows show possible movement directions. The boxed numbers indicate the numeric keys associated with the movements. (**C**) Task diagram: sequence of trial events. The green, red and blue tiles show the start, sub-goal and goal position, respectively. An example optimal trajectory is shown on the grid while using the KM from Figure 1B.

**Figure 2 brainsci-11-00292-f002:**
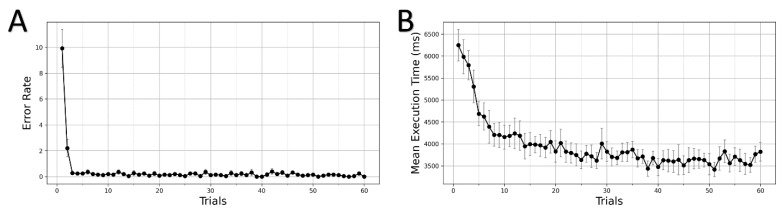
(**A**) Learning in grid-sailing task (GST): the error rates decrease as participants (*N* = 13) discover the optimal trajectory. (**B**) Trial-by-trial course of performance improvement in execution time across participants (*N* = 13) in successful trials. The bars on plot data-points denote standard error.

**Figure 3 brainsci-11-00292-f003:**
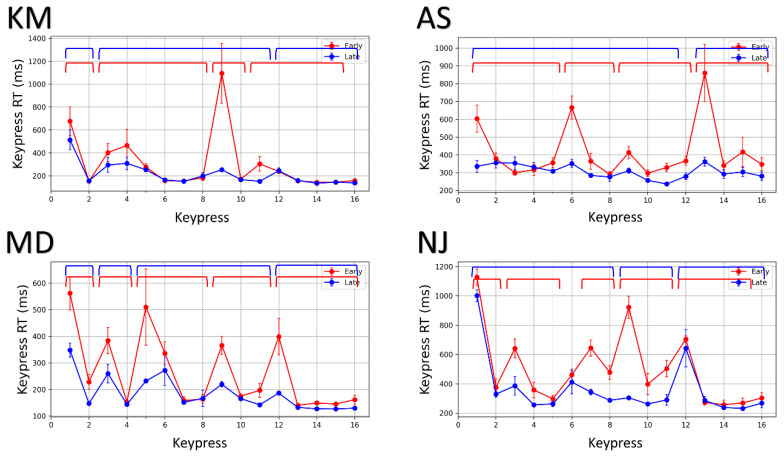
Re-organization of chunks with practice. Average keypress response time (RT) comparison plots for early (trials 1–10) and late (trials 50–60) phase in four representative subjects (K.M., A.S., M.D. and N.J.). The early and late phase RTs are plotted in red and blue, respectively. The brackets in red and blue on the top of each plot denote chunks in the early and late phases.

**Figure 4 brainsci-11-00292-f004:**
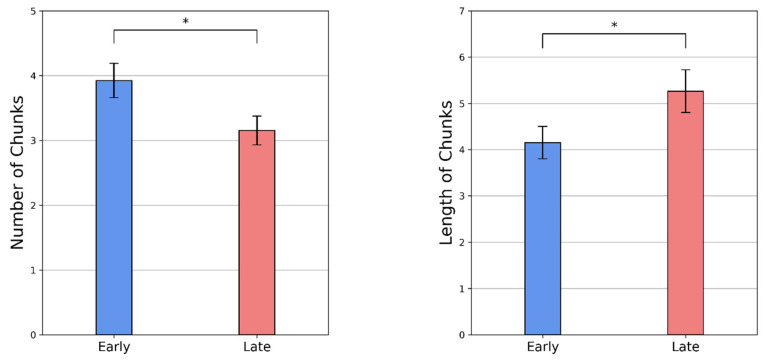
Evolution of chunking behavior with practice. The number of chunks significantly decreased, whereas the length of the chunks significantly increased from the early phase to the late phase. The bars denote standard error. * *p* < 0.05.

**Table 1 brainsci-11-00292-t001:** Re-organization of chunks with practice. A comparison of the number of chunks and length of chunks in the early and late phase for four representative subjects, K.M., A.S., M.D. and N.J.

SUBJECT	EARLY PHASE		LATE PHASE	
	No. of Chunks	Avg. Length of Chunks	No. of Chunks	Avg. Length of Chunks
K.M.	4	3.75	3	5.33
A.S.	4	4.00	2	7.50
M.D.	5	3.20	4	4.00
N.J.	5	2.80	3	5.33

## Data Availability

Not applicable.

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
