# Peer review of "Motor Chunking in Internally Guided Sequencing"

_brainsci, 2021, doi:10.3390/brainsci11030292_

Round 1

Reviewer 1 Report

Abstract

  • No comments

Introduction

  • The first paragraph requires more citations to set up the story for the reader (i.e., papers on motor skill learning). See below for suggested papers:
    • Flexibility in the development of action KE Adolph et al. The psychology of action 2, 399-426
    • Motor and language abilities from early to late toddlerhood: using formalized assessments to capture continuity and discontinuity in development A Ben-Sasson & SV Gill Research in developmental disabilities 35 (7), 1425-1432
  • The authors refer to memory load in the second paragraph on page 2. It would help readers if the authors included information on key factors that affect memory load such as dual tasking or the effects of biofeedback. See below for suggested papers.
    • Effect of DualTasking on Vibrotactile Feedback Guided Reaching - a Pilot Study. Shah VA et al. Haptics (2018). 2018 Jun;10893:3-14. doi: 10.1007/978-3-319-93445-7_1. Epub 2018 Jun 5.
    • Effects of biofeedback on control and generalization of nasalization in typical speakers ES Heller Murray et al. Journal of Speech, Language, and Hearing Research 59 (5), 1025-1034
    • Effects of singular and dual task constraints on motor skill variability in childhood SV Gill et al. Gait & posture 53, 121-126
  • In the authors’ description of internally guided motor skills, it is unclear whether they are always referring to motor tasks that are already learned (e.g., a piano piece that the musician plays from memory) or whether they are also including novel tasks in their definition. Please clarify.
  • Although the authors describe the Grid-Sailing Task, no rationale is provided about why this task is best to test their hypotheses. A sentence or two justifying this would be helpful.

Method

  • Please specify handedness for the 15 participants.
  • Please provide a rationale for why 9 seconds was used as the time to solve each trial.

Results

  • I understand why the authors only included successful trials in their analyses. However, I think that it would be interesting to at least have descriptive information about the error trials (e.g., percent of trials in which there were errors, how many errors, etc.). Also, were also 15 people included in the analyses, or were there some participants that had too many errors to include in the final analyses? On average, how many trials did each person contribute?
  • I’m assuming that Figure 2 is an aggregate of all subjects, but this is not clear. If this is the case, is this figure representative of how all subjects performed across trials?

Discussion

No comments

Author Response

We thank the reviewer for the suggestions and insightful comments. We have addressed all the concerns raised in a point-wise manner in the attached pdf file. We again thank you for your time and help with making the presentation better.

Reviewer 2 Report

The current study investigated the use of the Grid-Sailing Task (Fermin et al. 2010) to induce internally-guided sequence learning, and demonstrated that participants employ motor chunking as a strategy to complete the task, based on subject-specific temporal patterns of response time. The authors argue that this occurs without pre-specified or external-guidance and that the current model of the Grid-Sailing Task is a less constrained internally-guided sequencing model.

The study is well-written, however, several fundamental issues with choices of report, the statistical analysis employed, and unbalanced discussion points, must be addressed:

Major revisions

  • The introduction to the study needs careful consideration and revisions. While it is acknowledged that the framework for the current study should be highlighted, it seems overly comprehensive in the current state. This leads to difficulty in understanding which sections of your introduction refer to your hypotheses, and which are non-relevant to your main hypotheses and aim. Below, please find suggestions on how to improve the clarity of the message you are trying to convey
  1. Carefully revise your introduction so the importance of each section is clear. One suggestion could be to generally introduce (1) skill learning; (2) motor skill learning, including motor adaptation and motor chunking; (3) focus on motor chunking and its relation to sequence execution, memory load alleviation, and automatic control, as this is the primary parameter you are investigating; and (4) describe clearly why internally-guided motor chunking would represent a superior alternative to understand motor chunking in a real life setting (as opposed to externally-guided). This would make for a more clear and concise delivery of the importance of the current study, and why it is important to study. Avoid priority claims in your introduction unless proper citations are provided to support these statements.
  2. Ensure proper citation of underlying the framework you are presenting. In its current state, many of the paragraphs contain statements that are not validated by scientific literature.

  • There needs to be a better presentation of the subjects included in the study. For instance, the mean age should be presented with its associated standard deviation. Furthermore, it is not clear on which basis you included or excluded subjects in the study. Were there any other exclusion criteria involved? If so, these must be stated explicitly.

  • Was any sample size equation performed to estimate the number of participants needed for the study to demonstrate overall learning, organization of motor chunks, or efficient behavior? If so, explicit clarification is needed in the “Participants” section. If not, explicit reasons as to why not, must be provided.

  • Could the authors account for why a point system was introduced, when it is not described further in the results section? For instance, Fermin et al. 2010 fully accounted for all parameters associated with the grid-sailing task and would be highly relevant in relation to demonstrate how these relate to motor chunking. A full account for similarities and discrepancies with respect to the source material is encouraged in your discussion to fully reflect how your study stands out in comparison.

  • Following the above comment, reinforced learning involves reward as an integral part of the learning to drive future choices and correction of error, and it is unclear why reward was arbitrarily defined as success (=100) or no success (< 100). There needs to be a rationale for this methodological choice and how it affects the outcome of motor chunking. One could argue that even if only a score of 60 was achieved, this could still have been achieved using motor chunking.

  • It is unclear how the “trajectory” control is related to the task performance. If there is internally-guided sequencing in play, then having a locked “correct” and optimal path to the goal seems counterintuitive, since then, per definition it becomes externally-guided (i.e. they know the path they need to take). You describe this in your introduction “To complete the trial successfully, the participants can choose to reach the goal position using any possible optimal trajectories” and “The sequence of motor actions in such tasks is self-initiated (generated internally) by choice or internal plan (…) the characteristic difference between externally-guided and internally-guided sequencing is that, in the latter, the agent needs to plan or choose the sequence of motor actions by itself)”. Could the authors explain how this is achieved by setting up an optimal path that must be followed (unless ‘0’ score was to be rewarded in the end).

  • There are no available statistical analysis section. It is advised that this is implemented in the manuscript, in order to not only demonstrate knowledge on how to correctly analyse the data (given that you are looking into something that is not easily measurable). For instance, you provide a t-test result for early versus late phase, but the mean +- SD of the reaction times for the late phase = 257 +- 49, whereas the four included subjects you have represented have average lengths of chunks of 554 ms +- 144 ms. How is that representative of the group and not merely the hypotheses of the study itself? I would highly recommend an addition of a statistical section where you can share your thoughts on the included tests and why they are warranted in the way they are employed.

  • The results section is lacking detailed information on how you reached the conclusions you did. For instance, in section 3.2. Motor chunking in GST you write: “The improvement in execution time was observed even when participants followed an optimal trajectory. It suggests that sequence learning is not a mere by-product of error-driven learning”. I have a fundamental issue with this statement, since you cut-off success at =100 versus < 100. There could easily be noticeable change in execution time even if a score of 100 was not achieved – I suggest revising this to incorporate potential result bias given the arbitrary division of success versus no success.

  • With this in mind, a discussion on how memorizing a key sequence to ONLY reflect the optimal path (reward = 100) and how that relates to learning is warranted, since learning could still occur (as per definition in the reinforcement learning theory).

  • Graphically presenting the motor chunks does not necessarily mean that they are statistically different. There needs to be associated statistics supporting the statement “Figure 3 shows motor chunks in four representative subjects” and would highly recommend including group data instead of representative data as per my comment on reaction times not necessarily being representative of the group findings. Seeing the variability in the provided graphs opens up the question whether this particular pattern you are describing, is indeed representing the full sample.

Minor revisions

  • I would suggest to move the description of the grid-navigation tasks to the methods section, as this is not of importance to your aim and hypotheses, but a tool used to investigate your hypotheses.

Author Response

(The authors gave the same response as above.)

Round 2

Reviewer 2 Report

I have no further comments for the manuscript. I commend the authors for carefully interpreting and revising the manuscript to bring it to its current state. The paper now reads much more clear and I think you have done a great job in improving the message of the paper.

General comments: I think the Introduction now reads more easily in relation to the primary aim(s) of the study, however, for future reference I would still recommend to delineate the road from idea generation to hypothesis testing much faster. The amount of explanatory paraphrases still remains relatively high - and although this may be of personal preference, I still find it to obscure clarity. 

Presenting data with only representative subjects, while arguing that all subjects are along the same trend is counterintuitive, since you may as well show group-average data (which you essentially perform your statistics on, hence would be correct to visualize). This is of minor concern after having seen the supplementary, but I do believe it only shows part of the story.